# Traveling across Life Sciences with Acetophenone—A Simple Ketone That Has Special Multipurpose Missions

**DOI:** 10.3390/molecules28010370

**Published:** 2023-01-02

**Authors:** Fedor I. Zubkov, Vladimir V. Kouznetsov

**Affiliations:** 1Department of Organic Chemistry, Рeoples’ Friendship University of Russia (RUDN University), 6 Miklukho-Maklaya Street, 117198 Moscow, Russia; 2Laboratorio de Química Orgánica y Biomolecular, Escuela de Química, Universidad Industrial de Santander, Cl. 9 # Cra 27, A.A., Bucaramanga 680006, Colombia

**Keywords:** acetophenones, secondary metabolites, biological properties, allelochemical interactions, toxicity

## Abstract

Each metabolite, regardless of its molecular simplicity or complexity, has a mission or function in the organism biosynthesizing it. In this review, the biological, allelochemical, and chemical properties of acetophenone, as a metabolite involved in multiple interactions with various (mi-cro)organisms, are discussed. Further, the details of its biogenesis and chemical synthesis are provided, and the possibility of its application in different areas of life sciences, i.e., the status quo of acetophenone and its simple substituted analogs, is examined. In particular, natural and synthetic simple acetophenone derivatives are analyzed as promising agrochemicals and useful scaffolds for drug research and development.

## 1. Introduction

In a broad context, nature provides us with various “typical” life components, such as air, water, and food, enhancing our well-being and generously giving us the essentials for our survival. Reducing the focus to a scientific level, nature teaches us, sparks creativity, and improves problem-solving skills. One of the most valuable gifts from nature, i.e., from all living (micro)organisms, is diverse natural products with complex molecular architectures, sophisticated mechanisms of action, and extraordinary biopotentiality that can be converted into innovative leads or drugs [1,2]. However, each complex or simple molecule generated in nature acts as a metabolite for survival and reproduction, functioning as a small screw in the machinery of the biological systems of living organisms.

While primary metabolites have essential metabolic roles in nutrition and reproduction, secondary metabolites are nonessential to life but contribute to the suitability of the species for survival. Low-molecular-weight compounds, called small molecules, are often involved in defense mechanisms [3,4], i.e., the protection of vegetable or animal organisms against biotic or abiotic stresses, and are used as special chemicals, such as drugs, flavors, fragrances, insecticides, and dyes, by humans owing to their great economic value. In this minireview, one specific small molecule will be discussed—acetophenone (**1**) (Figure 1). Acetophenone has earned this honor due to its importance in chemical, ecological, biological, and medicinal areas of science owing to its extraordinary and irreplicable chemical, biological, pharmacological, and allelochemical properties.

The principal aim of this review is to briefly and conclusively answer the following questions: Why are we interested in this simple ketone? What is so special about acetophenone? Another objective related to these questions is to discuss acetophenone as a natural secondary metabolite or a semiochemical and a synthetic reagent/product to reveal its potential and prospects in core life sciences. The “destiny” of this small molecule is analyzed in this review, paying attention to young readers conducting biological, chemical, or medicinal studies on the single molecule and showing that even though acetophenone is simple and well known, it can lead to new and unexpected discoveries.

## 2. Historical Background

Acetophenone (CAS registry number 98-86-2), also known as methyl phenyl ketone, acetylbenzene, or hypnone, is the simplest of the ketones containing aromatic (phenyl) and aliphatic (methyl) groups. It is a colorless and flammable liquid (Bp. 201–202 °C) with a sweet, pungent odor. It is used as a fragrance ingredient in soaps, detergents, and perfumes and as an industrial solvent and valuable intermediate in the pharmaceutical and resin industries [5,6]. This aromatic ketone was found in the heavy-oil fraction of coal tar boiling at 160–190 °C. The first synthesis of acetophenone was conducted in 1857 by the French chemist Charles, who prepared the compound by treating benzoyl chloride with methyl zinc or by distilling a mixture of calcium benzoate and calcium acetate [7,8]. However, the first industrial synthesis of acetophenone, based on the reaction of benzene and acetic anhydride in an experiment conducted through the Friedel–Crafts reaction, was only realized in 1925 [6], 26 years after Friedel’s death. The physiological properties of acetophenone were investigated by Popof and Nencki; however, Beaumetz was the first to use it as a sleep-inducing agent, and hence, it was given the strange name hypnone [9,10]. Here, it is appropriate to cite the description of the odor of hypnone dated 1886: “Its odor is tenacious and very persistent, recalling at the same time oil of bitter almond and cherry laurel water” [10]. This is unlike the monotonous substance description used in modern times.

## 3. Acetophenone Is One of the Most Interesting Players in Skin Microbiota Manipulation by Vector-Borne Parasites Found in Animal-Feeding Insects

In July 2022, Cheng et al. reported that acetophenone released from the skin microbiota of flavivirus-infected hosts (mice and patients with dengue fever) acts as a potent attractant for *Aedes* mosquitoes, which are vectors for dengue (DENV2), chikungunya (CHIKV), and Zika (ZIKV) virus transmission, thereby increasing flavivirus transmission to aggravate dengue, chikungunya, and Zika diseases [11]. *Aedes* mosquitoes prefer to feed on mice infected by dengue and Zika viruses due to the released acetophenone compared to uninfected mice in control groups, in which acetophenone is absent. Female *Aedes aegypti* (*Ae. aegypti*) and *Aedes albopictus* (*Ae*. *albopictus*) mosquitoes were tested in these experiments, which showed that flaviviruses can stimulate the proliferation of acetophenone-producing skin commensal bacteria (especially the *Bacillus* genus) by suppressing the expression of the essential antimicrobial resistin-like molecule-α (RELMα) protein that protects against pathogenic bacterial skin infections, thereby killing the microbes. In particular, viruses can change host odors to attract mosquitoes.

It is worth noting that the use of isotretinoin, a naturally occurring retinoic acid and a vitamin A derivative, reduces attractiveness to mosquitoes through the activation of the expression of the RELMα protein in the skin of flavivirus-infected animals [11]. This interesting and important finding caught the attention of scientists who made brief but substantial and picturesque comments on Cheng’s work in prestigious journals such as *Science* [12], *Nature Reviews Microbiology* [13], and *Cell* [14]. Regardless, we wanted to examine this simple ketone from the perspective of synthetic and applied organic chemistry and chemical ecology fields.

It should be noted that the connection between the skin microbial population and attractiveness to mosquitoes and other blood-sucking insects has been studied, for example, in the African malaria mosquito *Anopheles gambiae* (*An. gambiae*) (infected with *Plasmodium falciparum* sporozoites), which plays an important role in malaria transmission [15,16,17,18], and the triatomine bug *Rhodnius prolixus*, which is the main vector transmitting the parasite *Trypanosoma cruzi*, the causative agent of Chagas disease [19,20]. Their host-seeking behavior is influenced by host scents, i.e., appropriate concentrations of volatile odors (a mixture of molecules, so-called molecular stimuli, such as ammonia, isobutylamine, acetone, lactic acid, isobutyric acid, and CO_2_) [19] present in human (or animal) skin microflora that can activate, attract, and even repel these blood-sucking insects.

Blood-sucking insects have excellent sensory abilities for detecting and following the physical and chemical signals (semiochemicals) emitted by their hosts; *Aedes* mosquitoes are not an exception. Cheng’s work [11] showed that the volatile acetophenone in the host skin microbiota produced a considerable electrophysiological response detectable by the antenna of *Ae. aegypti*, and thus, high levels of acetophenone attracted more mosquitoes than uninfected mice and healthy individuals in control groups. Nevertheless, close derivatives of acetophenone (**1**), 4-ethylacetophenone (**2**), and 4-ethoxyacetophenone (**3**) (Figure 1) had repellent effects on the malaria mosquito *An. gambiae* [18,21].

Moreover, acetophenone (**1**) and its *para*-ethyl analog **2** were found to exhibit significant repellency for the male Asian tiger mosquito, *Ae. albopictus*, but only at a high concentration (10^−2^) [22]. The latter metabolite has been described as a male-swarming aggregation pheromone for *Ae. aegypti* mosquitoes [23], which increases female attraction and mating success. However, none of the three acetophenone derivatives **1**–**3** were identified among specific semiochemicals, i.e., acetoin (3-hydroxy-2-butanone), sulcatone (6-methyl-5-hepten-2-one), octanal, nonanal, and decanal, released from the males of *Anopheline* species—*Anopheles arabiensis* and *An. gambiae* [24]. Such pheromones can remarkably manipulate these highly dangerous insects.

On the other hand, acetophenone (**1**) was found to be an effective repellent in field experiments [25] for tsetse flies (*Glossina* spp.) infected by *Glossina hytrosavirus*, the primary vector of African trypanosomes, which cause human African trypanosomosis (HAT or sleeping sickness) and African animal trypanosomosis (AAT or nagana) [26]. Recently, it was reported that acetophenone-containing Zebra skin odor repels the savannah tsetse fly [27,28], confirming previous results from field experiments [25]. Therefore, the diametrically opposite bioeffects of acetophenone (**1**) mentioned above could be used in the control and prevention of heralded diseases. This volatile ketone could be a suitable biomarker for detecting pathogens that actively manipulate host odors. However, the development of these volatile-based diagnostics, which use the “signal” of disease and the background “noise” of genetic and environmental variations, is still in its infancy [29,30]. On the other hand, odor-bait technology as a surveillance and control tool for insect vectors, such as tsetse flies or *Aedes* mosquitoes, has been promoted as a new and viable component of the integrated vector management program [31,32,33]. In this context, some extracts and essential oils from aromatic plants could be more advantageous in the biological control of pest insects. Nonetheless, there is some skepticism over this aspect [34].

Another analog of acetophenone, i.e., 2-aminoacetophenone, secreted by *Pseudomonas aeruginosa*, which is a ubiquitous opportunistic human pathogen, can facilitate attraction to food for several fly species, including *Musca domestica*, *Ceratitis capitata*, and *Drosophila melanogaster* [35].

## 4. Acetophenone Is a Prolific Semiochemical for Plant-Feeding Insects in Complex Interspecific Communication

Investigations on the involvement of plant volatiles in the feeding attraction behavior of mosquitoes were conducted before Chang’s work [11]. Mosquitoes visit flowers for nectar and may, in turn, act as pollinators for plants [36]. Like animal skin microflora, the composition of the inflorescence odor of plants can be an attractant or a repellent for mosquitoes [37].

Gonzalez-Audino et al. demonstrated that freshly cut inflorescences of the perennial herb “alyssum” *Lobularia maritima* (Cruciferae) stimulated a positive flight response in both sexes of *Ae. Aegypti* [38]. Moreover, acetophenone and four other compounds (1-octanol, 2-phenylethanol, benzyl cyanide, and benzyl isothiocyanate) were identified from the headspace of *Lobularia maritima* (*L. maritima*) by direct comparison with analytical standards. By testing the flight preference toward single synthetic compounds identified in the headspace of *L. maritima*, the authors showed that acetophenone (**1**) was the first plant-derived compound that elicited positive flight behavior in *Ae. aegypti* (Figure 2).

Note that this volatile compound is present in the whole plant, which is likewise attractive to the female *Diachasmimorpha longicaudata* wasp, which is a solitary larval endoparasitoid of Caribbean fruit flies (*Anastrepha suspensa*) and a major fruit pest that occurs in the New World tropics and subtropics [39]; however, while the males of this species can recognize ketone **1**, it did not elicit a behavioral response [40]. Volatiles of *Alyssum* selectively attract and enhance the performance of another larval parasitoid (*Cotesia vestalis*) of the diamondback moth (*Plutella xylostella*), which is one of the most important pests in cruciferous plants worldwide [41]. Therefore, this acetophenone-containing plant, rich in nectar, which is the main sugar source for mosquitoes, could be a good candidate for use as a trap crop for these pest insects, achieving successful biological control.

Regarding coniferophagous bark beetles of the *Dendroctonus* genus, important damaging insect pests of pines, acetophenone (**1**) was identified in the volatiles of females of three species—*Dendroctonus pseudotsugae* (*D. pseudotsugae*), *Dendroctonus ponderosae*, and *Dendroctonus rufipennis*. However, it was found that acetophenone reduced only the response of female *D. pseudotsugae* to the attractive bait. This suggests that acetophenone (**1**) could serve as a repellent kairomone during host selection [42]. Other species, *Dendroctonus frontalis* [43] and *Dendroctonus brevicomis* [44,45,46], also produce ketone **1**, described by the authors as an antiattractant compound that inhibits the attraction of conspecifics. Interestingly, these aggressive tree-killing bark beetles can rapidly colonize their host by producing an aggregation pheromone, which can be regulated by acetophenone-based multicomponent blends. It is interesting to note that other pest beetles are addicted to acetophenone, which had a positive impact on the weight and multiplication of the red flour beetle (*Tribolium castaneum*), a worldwide pest of stored products, and the tobacco beetle (*Lasioderma serricorne*), one of the most widespread and damaging pests for the tobacco industry, when they were treated with acetophenone-containing cultures that deter the beetles from feeding well, which affected their growth, and thus, acetophenone exhibits antifeedant properties in this case [47]. Acetophenone is also involved in complex interactions between fungal symbionts of the ambrosia beetle *Platypus cylindrus* and the cork oak (*Quercus suber*), an interesting plant that is the primary source of cork for wine bottle stoppers. These interactions are linked to the cork oak decline process. It was found recently that strains of fungal associates of *P. cylindrus*, which is a major cork oak pest in Portugal, induced an increase in volatiles known for mediating ambrosia beetle behavior (mainly acetophenone, sulcatone, and nonanal) in response to fungal inoculation. This may help to elucidate the mutualistic or pathogenic aspects of these complex symbiotic interactions and develop new control strategies for *P. cylindrus* [48].

The floral scent of *Antirrhinum majus pseudomajus* (*A. majus pseudomajus*), a wild subspecies of snapdragon, also contains acetophenone as the main component, contributing over 69% of the absolute emissions of this plant. Suchet et al. showed that bumblebees (*Bombus terrestris*), flower visitors in the *Antirrhinum* genus, innately avoided *A. majus pseudomajus*, taking care of their fragile existence. Therefore, acetophenone (**1**) again turned out to be a repellent compound [49]. It seems that 4-ethylacetophenone (**2**), present as a minor constituent in the peels of navel oranges (*Citrus sinensis*), could also be involved in complex interrelationships with the Mediterranean fruit fly, *Ceratitis capitata*, eliciting a significant voltage spike in females but not in males (Figure 2) [50].

Moreover, acetophenone is the major component (51%) of the volatile oil of *Pogostemon heyneanus* (Indian patchouli) leaves [51], which could be useful for odor-bait technology. This oil also showed antibiofilm and antivirulence effects against *Streptococcus pyogenes*, an important human pathogen that causes several superficial infections and invasive diseases [52].

## 5. Biogenesis: How Can Acetophenone Be Formed In Vivo?

Acetophenone is one of the secondary metabolites and could be classified as a phenolic C_6_–C_2_ metabolite according to its basic skeleton. As mentioned previously in the Historical Background section, acetophenone is a volatile organic compound and a polar, lipophilic liquid (ε = 17.40; μ = 3.05 D; logP = 1.58) with a low molecular weight (MW = 120.15 g·mol^−1^) [53,54]. Its physical properties allow acetophenone to freely cross cellular membranes and be released into the surrounding environment.

Phenolic compounds are ubiquitously present in the plant kingdom but are less common in bacteria, fungi, and algae [55]. Generally, plant phenolics are believed to act as defense compounds against herbivores, microbes, viruses, and compelling plants or as signal compounds to attract pollinating or repel seed-dispersing animals, mainly insects. They are mainly synthesized via the shikimate pathway [56,57], which is present in bacteria, fungi, and plants but is absent in animals. It provides phenylalanine (Phe) and tyrosine (Tyr), precursors of the corresponding cinnamic derivatives. In particular, Phe is obtained from the conversion of phosphoenolpyruvate (PEP) via the glycolysis pathway and erythrose 4-phosphate (E4-P) via the pentose phosphate pathway into chorismite via 3-deoxy-D-arabino-heptulosonate-7-phosphate (DAHP) through DAHP synthase (DAHPS; EC 2.5.1.54). Furthermore, the well-known and widely distributed enzyme L-phenylalanine ammonia lyase (PAL; EC 4.3.1.5) deaminates Phe to *trans*-cinnamic acid (CA) [58] (Figure 3).

Notably, Phe biosynthesis occurs in plastids, and its further conversion to volatile compounds, such as acetophenone, occurs outside this organelle, especially in the peroxisome, a small membrane-enclosed oxidative vesicle. The detailed bio-origins of acetophenone are still unclear [56,57,58]. It is known that *trans*-CA can subsequently be hydroxylated to β-hydroxy-phenyl propionic acid (HPPA) and then to β-oxo-phenyl propionic acid (OPPA), which can be degraded into acetophenone. This biosynthetic pathway, called the β-oxidative pathway [59], was proposed for the fungus *Bjerkandera adusta* by Lapadatescu et al. [60] and explored by Dong et al. in the flowers of tea plants (*C. sinensis*) [61]. Notably, although the enzymes involved in these reactions have not yet been identified, this natural route to acetophenone is a very significant model to follow and could help develop new industrial methods for this valuable ketone based on nonpetrol sources.

## 6. Industrial Production of Acetophenone: So Many Possibilities, but None Are Ideal and Sustainable

Contrary to the biogenic process, industrial methods for producing acetophenone are based on petroleum derivatives, an exhaustible resource. Almost 50 years ago, Sanders et al. [6] appropriately analyzed eight possible syntheses of acetophenone (**1**) that could attain commercial status. Among them, the acid-catalyzed Friedel–Crafts acetylation of benzene (**4**) with acetic anhydride or acetyl chloride (method A) [62], the oxidation process of cumene (**5**), known as the Hock process (for phenol and acetone production), based on the decomposition of cumene hydroperoxide (**6**) under acid catalysis conditions [63,64,65] (method B), and the catalytic liquid-phase oxidation of ethylbenzene (**7**) with oxygen from air [66], giving the required acetophenone as the main product (together with 1-phenylethanol) through ethylbenzene hydroperoxide (**8**) (method C), are still the main procedures in the modern acetophenone industry (Figure 4).

First, using the Hock process, acetophenone is formed as a byproduct, and it occurs after the neutralization (using NaOH) of excess acid. Second, the ethylbenzene oxidation reaction can be conducted industrially under catalyst- and solvent-free conditions at high temperatures (141–148 °C) and 3 atm pressure [67,68]. Finally, during the aerobic oxidation reaction, ethylbenzene conversion is maintained at ~12 wt% [69].

The Hock process (method B) dominates the synthesis of acetophenone, with over 90% acetophenone formed by this process, although acetophenone is formed as a byproduct in this process. The acid-catalyzed Friedel–Crafts acetylation of benzene (method A) and the oxidation of ethylbenzene (method C) make up the rest. Notably, the global market for this ketone was valued at USD 219 million in 2021, and by 2022, the market is projected to be valued at USD 225 million [70,71]. At first glance, the modern acetophenone industry is a model of success and triumph. However, the urgent need for new, effective, and sustainable methods is clearly seen because all current methods suffer from numerous serious shortcomings, mainly the use of excess AlCl_3_, which causes the severe corrosion of equipment and the contamination of products, resulting in acidic wastewater after the additional cleaning process [6,72] (methods A–C).

Other disadvantages include polyethylbenzene formation, i.e., the transalkylation process; low selectivity for forming acetophenone; working under high-pressure conditions; complicated reactions due to used catalysts; complicated handling; and the corrosive and environmentally unfriendly nature of H_2_SO_4_ (method C). However, the latter process is considered to be a more promising route to acetophenone, and thus, it is the object of intensive studies for efficient, environmentally friendly, and low-cost heterogeneous routes for acetophenone production [73,74].

Additionally, the homogeneous liquid-phase Mn(OAc)_2_-CoBr_2_ (MC-system)-catalyzed oxidation of ethylbenzene in the presence of air and AcOH to acetophenone in a continuous flow mode has been developed [75]. Notably, the 1-phenylethanol byproduct is also oxidized to acetophenone under reaction conditions, allowing an acetophenone selectivity of ~74% at an ethylbenezene conversion rate of ~96% after 150 min at 80 °C. This continuous processing of acetophenone is clearly advantageous from economic and environmental perspectives because it reduces reaction times, increases volume productivity, and avoids byproduct formation.

## 7. Natural and Synthetic Closely Related Acetophenone Cousins as Promising Agrochemicals

Both plant metabolism and chemical synthesis provide us with the molecular acetophenone family, an interesting group of alkyl-phenylketones that have an aromatic homomonocyclic framework. Owing to its chemical nature and reactivity, acetophenone, the parent of this family, is one of the most useful precursors in heterocyclic synthesis [76,77,78] and the ideal synthon for multicomponent reactions based on simple aldol condensations or the α-functionalization strategy [79], allowing the preparation of various natural product analogs and pharmaceuticals with important biological properties.

On the contrary, biogenetic acetophenones are pervasive in the vegetal world and are catabolized by microorganisms [60,80]. As discussed above, some plants (both angiosperms and gymnosperms) produce acetophenone derivatives (e.g., hydroxy acetophenones) to protect themselves from insects, which are conjugated to glucose (hydroxy acetophenone glucosides) and cleaved upon insect attack [81,82], which shows that acetophenones are inherently involved in biological systems. Among endless, structurally different acetophenone derivatives of natural or synthetic origin, our attention was drawn to the selected group of mono-, di-, or trisubstituted acetophenone molecules with general formula **A** (Figure 5). These derivatives are usually commercially or/and synthetically available inexpensive reagents and generally show low toxicity in eukaryotic cells (e.g., the oral LD_50_ for acetophenone in the rat varies from 900 to 3200 mg/kg [83]).

Based on these facts, it can be assumed that acetophenones are interesting and promising models in agricultural and pharmacological research. As agricultural crops are facing enormous losses due to pest attacks, diseases, and weed damage, which result in direct economic losses, including a decrease in grain yield and quality, the research and development of new pesticides is an ongoing and important task in spite of negative public opinions on pesticide use, which are due to the present hazards of pesticides (health, environmental, and resistance problems) [84]. Thus, there is clear evidence that pesticides will continue to be a vital product in a diverse range of technologies that can maintain and improve living standards for people globally [85,86]. Notably, without the use of pesticides, there would be a 78% loss in fruit production, a 54% loss in vegetable production, and a 32% loss in cereal production [87]. However, current studies on pesticides are focused on reexamining natural products, e.g., for the structural development of secondary metabolites, as an eco-friendly possible alternative to synthetic pesticides.

The usage of so-called biopesticides has become increasingly popular in recent years, and they are considered safer than conventional pesticides [88]. In this context, simple acetophenone derivative **A** as a potential pesticide could be a useful model for new biopesticides, considering that acetophenone is the second most abundant component of Ridomil^®^, which is a very common fungicide formulation used in horticulture and vineyards [89]. Hydroxy-substituted acetophenones **9**–**17** (Figure 5) exhibit considerable antifungal activity against important fungal plant pathogens of the genera *Colletotrichum*, *Botrytis*, *Alternaria*, and *Fusarium* and could be useful models for controlling these agricultural diseases [90,91,92,93,94].

Among synthetic acetophenone ethers **9**–**11**, compound **10** bearing a prop-2-ynyloxy group was found to be active against seven phytopathogenic fungi, i.e., *Colletotrichum gloeosporioides*, *Botrytis cinerea*, *Alternaria solani*, *Fusarium oxysporum* f. sp. *vasinfectum*, *F. oxysporum* f. sp. *niveum*, *F. solani* var. *coeruleum*, and *F. graminearum*, exhibiting good inhibitory concentration (IC_50_) values ranging from 37 to 87 μg/mL [90]. In contrast, natural hydroxy-acetophenone allelochemicals **12**–**15** displayed poor activity against *Cytospora* sp., *Glomerella cingulata* Schr, *Pyricularia oryzaecar*, *Botrytis cinerea* Pers et Tris, and *Alternaria solani*. A simple replacement of the acetyl group with the *iso*-butyryl fragment in the phenyl ring of **15** considerably improved antiphytopathogenic activity; 1-(2,4-dihydroxy-5-methylphenyl)-2-methylpropan-1-one (**16**) exhibited substantial activity (IC_50_ = 17–37 μg/mL) [92], and its C-4-methoxy analog **17** was one of the most active acetophenone derivatives (IC_50_ = 0.9–19.5 μg/mL) [93] and might be a very promising candidate for developing new antifungal agrochemicals. Compounds **14** and **15** can act as inhibitors of class II fructose-1,6-bisphosphate aldolase, an enzyme critical for bacterial, fungal, and protozoan glycolysis/gluconeogenesis [94].

Plant diseases are caused not only by fungal and bacterial pathogens but also by numerous insects and endoparasitic animals, especially, plant-parasitic nematodes. The latter can be categorized as helminths (small, microscopic roundworms). Among them, root-knot nematodes (especially *Meloidogyne* spp.) are one of the major pests of economic importance in vegetable crop production globally [95]. Halogen (nitro)-substituted acetophenones **18**–**25** (Figure 5) were found to be very potent in inducing the paralysis and death of the root-knot nematode *M. incognita* [96,97] in susceptible tomato plants (*Solanum lycopersicum* L.; cv. Rutgers) after 24 h (EC_50_ = 2.5 and 54.8 mg/L) as well as after 72 h (EC_50_ = 2.3 and 65.6 mg/L). Among these, 2,4-dichloroacetophenone (**20**) was found to be the best nematicidal agent, exhibiting good inhibition concentration parameters (EC_50_/24h = 2.5 ± 13.7 mg/L and EC_50_/72h = 2.3 ± 5.5 mg/L), and thus could be used for developing new commercial ingredients [96]. On the contrary, it should be noted that 2,4-dihydroxyacetophenone (**13**) from the hydroxy-substituted acetophenone series showed good nematicidal activity in second-stage juveniles (J2) of *M. incognita* in vitro, increasing the following lethal concentration of J2 to 50% [(LC_50_) = 210 μg/mL] compared with that of carbofuran (150 μg/mL) [98], one of the most toxic carbamate pesticides. Thus, this acetophenone derivative could also be used to formulate new commercial ingredients.

Finally, herbicides are the most widely used type of pesticide, as weeds are a major constraint that limits yield in many crops. Herbicides represent ~50% of all crop protection chemicals used globally, compared to insecticides and fungicides, which are ~17% each [87]. Several studies have found that ketones, including substituted acetophenones **26**–**29** (Figure 5), are good herbicides. Methyl phenyl ketone allelochemicals from plants, such as xanthoxyline (**26**) [99,100] and acetosyringone (**27**) [101] (Figure 5), are suitable models for developing potential herbicides. Both acetophenone derivatives displayed potent inhibitory effects on the shoot growth of barnyard grass (*Echinochloa crus-galli* (L.) Beauv.) under laboratory conditions.

Furthermore, xanthoxyline showed a significant inhibitory effect on seed germination (15%) and inhibited the root growth (63%) of this annual grassy weed. However, acetophenone (**1**) and most of its *para*-monosubstituted derivatives **12** and **18**–**24** showed weak activity at a concentration of 400 μM on barnyard grass and Chinese amaranth (*Amaranthus tricolor* L.) [102]. Regarding the *ortho*-monosubstituted acetophenone series, there is little information on its biological properties; however, the remarkable and specific seed germination inhibition effect (~75%, Chinese amaranth) of *o*-nitroacetophenone (**25**) allows us to consider it for the molecular design of new acetophenones that are active against weeds. Notably, molecular docking on the 4-hydroxyphenylpyruvate dioxygenase (HPPD) enzyme indicated that these acetophenone derivatives may be involved in key interactions of HPPD inhibitors [102]. This finding is useful for developing small ketone herbicides.

## 8. Acetophenone Skeleton for Developing Pharmacological Agents/Drugs

Acetophenone, with an LD_50_ value of 815 mg/kg (oral, rats), is classified as a group D carcinogen (which means it is not a human carcinogen) and is also used in the pharmaceutical industry for several purposes. First, some of the developed pharmacological agents containing the acetophenone moiety have been proposed for practical use in the treatment of different diseases, for example, α-aminoketone drugs, the antidepressant medication bupropion (**30**) [103], the anorectic drug amfepramone (diethylpropion, **31**) [104], and the neuropsychiatric agent pyrovalerone (**32**) [105] (Figure 6). Both so-called synthetic cathinones, amfepramone and pyrovalerone, which are chemically similar to amphetamines, have been proposed as appetite suppressants, although they are not currently in clinical use.

Second, acetophenone derivatives are also one of the most valuable precursors in drug synthesis. The hypnotic–sedative agent zolpidem (**33**) [106], the calcimimetic agent cinacalcet (**34**) [107], and the topical antifungal agent oxiconazole (**35**) [108] (Figure 6) are some examples of the drugs synthesized using acetophenone derivatives.

Finally, simple, natural, and synthetically substituted acetophenones can be an ideal model or prototype in drug research and development, for example, hydroxy acetophenones such as paeonol (2-hydroxy-4-methoxyacetophenone, PA, **36**) and apocynin (AP), also known as acetovanillone (4-hydroxy-3-methoxyacetophenone, **14**). Both are plant-derived compounds displaying anti-inflammatory properties without any side effects, even after long-term administration. The former is a major constituent of Cortex Moutan [109], known as tree peony root bark (*Paeonia suffruticosa*), and has been used in traditional Chinese medicine for more than 1000 years. It is also present in *Dioscorea japonica* [110], known as the East Asian mountain yam, and *Paeonia clusii* [111].

Besides its anti-inflammatory properties [112], PA exhibits several interesting biological properties, such as analgesic, antioxidant, anticancer, neuroprotective, cardioprotective, and antidiabetic properties, among others [113,114]. PA’s interesting and promising partner, AP, was first identified as a biologically active substance in the roots of *Apocynum cannabinum* (Canadian hemp), an herbaceous plant [115,116], later in the roots of *Picrorhiza kurroa* Royle ex Benth, a perennial plant growing in the alpine Himalaya [117,118], and then in *Ziziphora clinopodioides*, which is used in traditional Uighur medicine for many purposes [119,120].

More specifically, AP acts as a strong inhibitor of the production of reactive oxygen species (ROS) by activating human polymorphonuclear neutrophils (PMNs). Because PMNs and ROS play a central role in the innate host defense against invading microorganisms, the activity of AP could be vital for treating diseases with neutrophils as (pro)inflammatory mediators and might be useful for treating the neuroinflammatory component of neurodegenerative diseases [121,122]. PA can trigger the innate immune system by affecting Toll-like receptor 4 (TLR4), which majorly influences the inflammatory signaling pathways of the intestinal tract; inhibits the release of proinflammatory cytokines induced by the nuclear factor (NF-κB) mitogen-activated protein kinase (MAPK); and suppresses the expression of inducible nitric oxide synthase (iNOS), a key enzyme generating nitric oxide from L-arginine [123,124,125].

Therefore, the combination of both hydroxyacetophenone isomers (APPA, **14/36**) can have additional beneficial effects on inflammatory diseases, such as rheumatoid arthritis [126] or osteoarthritis [127], in which neutrophils and TNFα signaling are important for pathologies. Indeed, in animal models, APPA treatment has demonstrated improvements in pain and function during the management of osteoarthritis in dogs, with comparable effects to nonsteroidal anti-inflammatory drugs (NSAIDs) such as meloxicam [128], and APPA is now being developed by AKL Research and Development as a medication for treating osteoarthritis [129,130].

However, similar to hydroxy acetophenones **14** and **36**, APPA has poor solubility and bioavailability, which hinders its development as a pharmaceutical product. Thus, the development of a sure-fire delivery system for the APPA combination with enhanced efficacy and bioavailability still needs further research [114].

Trihydroxyacetophene derivatives show also various interesting pharmacological activities and serve as excellent precursors for bioactive substituted chalcones. Among them, 2,4,6-trihydroxy-3-geranyl acetophenone isolated from the medicinal plant *Melicope ptelefolia* [131] demonstrated significant pharmacological activities against inflammation, endothelial and epithelial barrier dysfunctions, asthma, allergies, and cancer through its modulatory actions on specific molecular targets, being safe for consumption and easily available to be prepared synthetically [132,133]. Therefore, all of this also positions it as a drug lead in the current pharmaceutical industry, like APPA.

## 9. Conclusions

Secondary metabolites such as acetophenone and its simple analogs not only perform survival functions for the organisms synthesizing them but also provide us with tips for improving our lives. Our task is to better understand the complex intra- and interspecific interactions in which these molecules participate. When we succeed, we can create new, versatile, and effective models based on acetophenones that can be used for different human activities without damaging our surroundings and the organisms that generate them.

Acetophenone is one of the most exciting players in skin microbiota manipulation by vector-borne parasites from animal-feeding insects. Acting as an allelochemical agent, acetophenone and its simple derivatives can be an attractant or a repellent for blood-sucking insects, i.e., mosquitoes, flies, and ticks, vectors of dangerous tropical infectious diseases, including malaria, sleeping sickness, Chagas disease, chikungunya, dengue, and Zika virus. Both properties can be used to our benefit to safeguard human lives.

Like animal skin microflora, the composition of the inflorescence odor of nectar-rich acetophenone-containing plants can be attractive and repellent for insects, which are the most important pests of economically important crops. Therefore, these plants could be good candidates for use as trap crops for these pest insects, achieving successful biological control. Some simple acetophenones serve well as an inspiration for novel agrochemicals. There are different types of pesticides, such as fungicides, insecticides, and herbicides. With the appropriate combination of acetophenone derivatives, it is possible to formulate new commercial ingredients for sustainable agrochemicals.

The development of pharmacological agents/drugs should also be considered. Acetophenone-based prototypes inspired by traditional medicines are very suitable and promising in drug research. The inspiring example of APPA development teaches us that the lessons of nature must be analyzed in detail. We hope this example will not be the last. There are many models in nature that are yet to be discovered. A caring attitude toward nature as a teacher will give us the opportunity to grow without destroying it.

We hope that the information provided in this article will encourage future research on single secondary metabolites, such as acetophenone, from a multi-focus point of view. Only in this way can we uncover the true meaning of each metabolite.

## Figures and Tables

**Figure 1 molecules-28-00370-f001:**
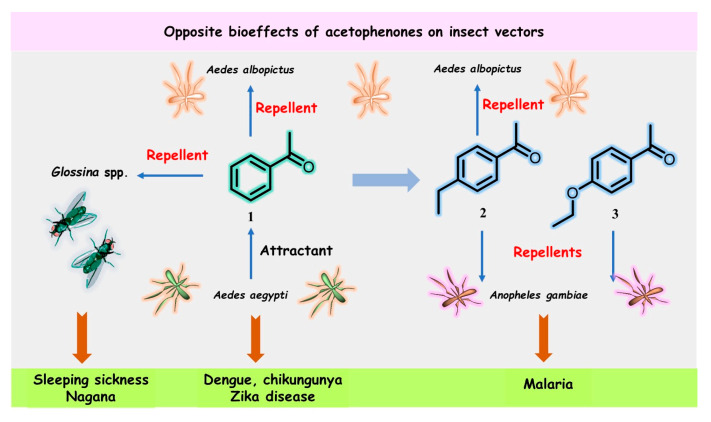
Structures and opposite bioeffects of acetophenone (**1**) and its derivatives **2** and **3** on insect vectors of dangerous tropical infectious diseases, including malaria, sleeping sickness, chikungunya, dengue, and Zika virus, transmitted by mosquitoes, flies, or ticks.

**Figure 2 molecules-28-00370-f002:**
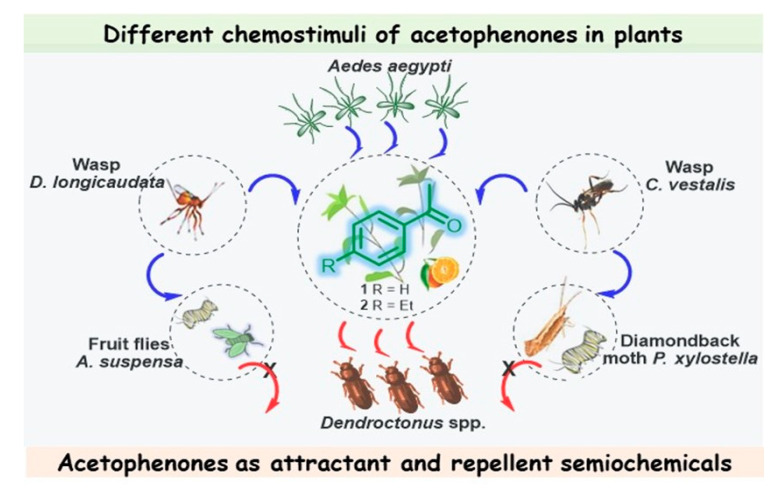
Different chemostimuli of acetophenones **1** and **2** in *L. maritima* and *Citrus sinensis*; multitrophic interactions between plants, herbivores, and their natural enemies, in which acetophenones act as attractant and repellent semiochemicals.

**Figure 3 molecules-28-00370-f003:**
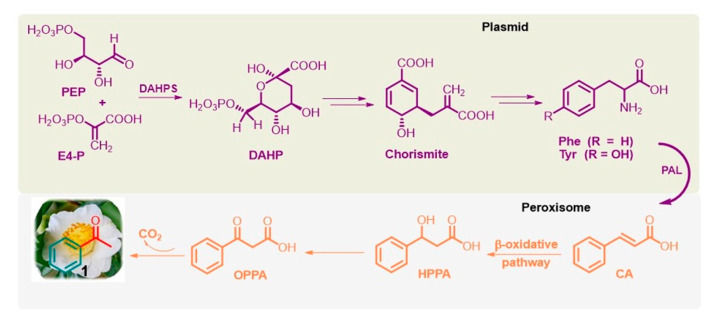
Simplified shikimate pathway of L-Phe from PEP and E4-P, and the β-oxidative pathway of acetophenone (**1**) from CA in flowers of tea plants and fungus *Bjerkandera adusta*.

**Figure 4 molecules-28-00370-f004:**
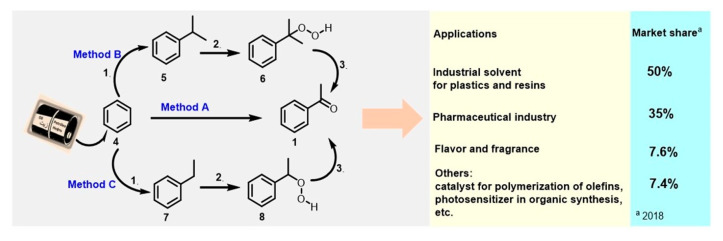
Industrial production of acetophenone and its applications. Method A. Ac_2_O or MeCOCl/AlCl_3_. Method B. Step 1: propylene/AlCl_3_-HCl or kieselguhr/H_3_PO_4_; Step 2: air/oxygen as an oxidant/NH_4_OH; Step 3: acid-catalyzed cleavage with H_2_SO_4_ and then NaOH treatment. Method C. Step 1: ethylene/AlCl_3_; Steps 2 and 3: air/oxygen as an oxidant/Co(II) and Mn(II) catalysts in acetic acid media.

**Figure 5 molecules-28-00370-f005:**
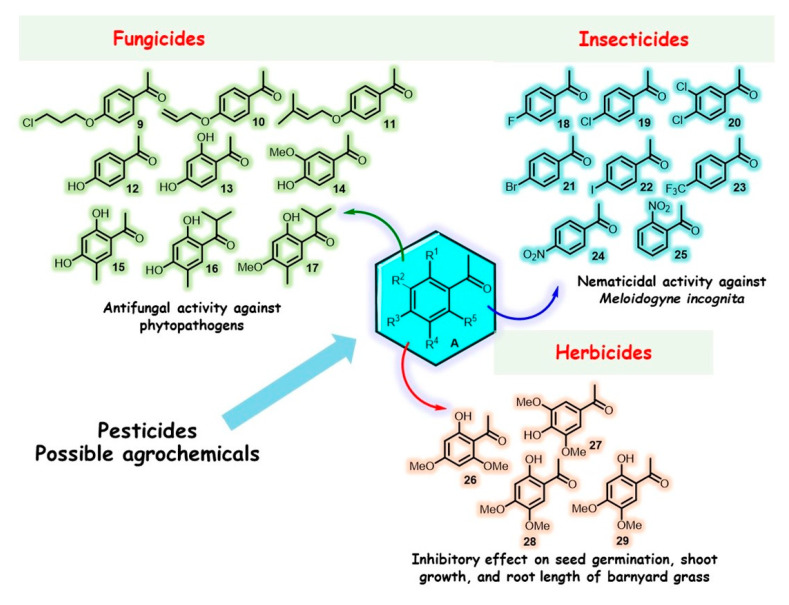
Structures of selected mono-, di-, or trisubstituted acetophenones as promising models in agricultural research for fungicide, insecticide, and herbicide formulations.

**Figure 6 molecules-28-00370-f006:**
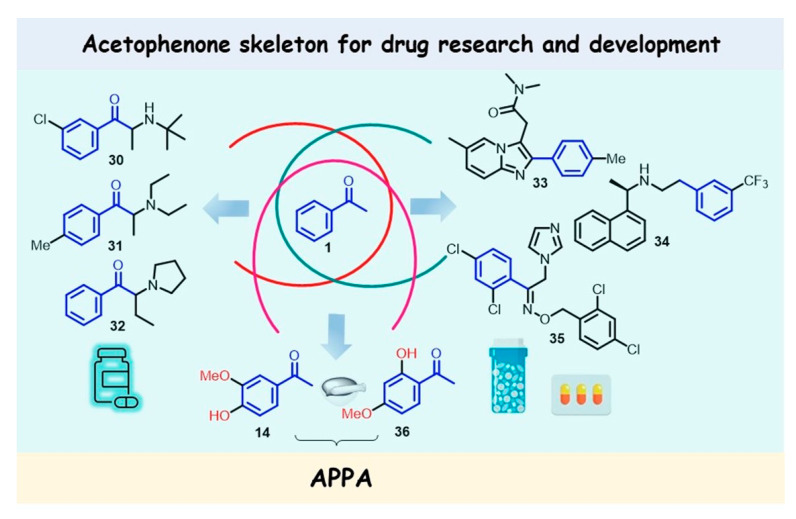
Structures of α-aminoacetophenone drugs: bupropion (**30**), amfepramone (**31**), and pyrovalerone (**32**) and selected drugs zolpidem (**33**), cinacalcet (**34**), and oxiconazole (**35**), which are prepared using simple acetophenone derivatives as starting materials and the combination of both hydroxyacetophenone isomers (APPA, **14**/**36**).

## Data Availability

Not applicable.

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
