# Peer review of "Traveling across Life Sciences with Acetophenone—A Simple Ketone That Has Special Multipurpose Missions"

_molecules, 2023, doi:10.3390/molecules28010370_

Round 1

Reviewer 1 Report

In manuscript “Traveling across life sciences with acetophenone – a simple ketone that has special multipurpose missions,” authors provide an interesting view about some biological, and allelochemical properties of acetophenone, as well as some important synthetic approaches to this compound. I recommend the publication of this review in Molecules after an additional revision.

Minor typographical mistakes are found, for example:

-Page 6, line 231 “et al.6 …” reference is in superscript.

Reviewer 2 Report

In my view, this review manuscript is very interesting and also important for the scientists who work in the field of VOCs. The authors are familiar with this simple metabolite and should have read enough publications on the chemical. The language and the organization of the MS are very nice. I hope it will soon be published in the journal, which will provide chance for readers to learn this VOC compound.  

Although most of the updated publications have been included, there are a few flaws before being accepted for publication.

1. In section 3, there are a few other acetophenone VOCs functioning in animal-feeding insects, please supply such works. The list references are associated with the topic.

(1) Jonfia-Essien WA, Alderson PG, Tucker G, Linforth R, West G. The growth of Tribolium castaneum (Herbst) and Lasioderma serricorne (Fabricius) on feed media dosed with flavour volatiles found in dry cocoa beans. Pak J Biol Sci. 2007; 10(8):1301-4. doi: 10.3923/pjbs.2007.1301.1304. PMID: 19069932.

(2) Kapsetaki SE, Tzelepis I, Avgousti K, Livadaras I, Garantonakis N, Varikou K, Apidianakis Y. The bacterial metabolite 2-aminoacetophenone promotes association of pathogenic bacteria with flies. Nat Commun. 2014; 5: 4401. doi: 10.1038/ncomms5401. PMID: 25043228.

(3) Nones S, Sousa E, Holighaus G. Symbiotic Fungi of an Ambrosia Beetle Alter the Volatile Bouquet of Cork Oak Seedlings. Phytopathology. 2022; 112(9):1965-1978. doi: 10.1094/PHYTO-08-21-0345-R. Epub 2022 Aug 2. PMID: 35357159.

2. In the section 4, the authors should supply some contents related and published publicly such as the following article: 

Jonfia-Essien WA, Alderson PG, Tucker G, Linforth R, West G. Behavioural responses of Tribolium castaneum (Herbst) to volatiles identified from dry cocoa beans. Pak J Biol Sci. 2007; 10(20):3549-56. doi: 10.3923/pjbs.2007.3549.3556. PMID: 19093461.

3. In the section 5, you also need to supply some, such as the following publication

Li H, Sasmal A, Shi X, Soulé JF, Doucet H. Halo-substituted benzenesulfonyls and benzenesulfinates: convenient sources of arenes in metal-catalyzed C-C bond formation reactions for the straightforward access to halo-substituted arenes. Org Biomol Chem. 2018; 16(24):4399-4423. doi: 10.1039/c8ob00632f. PMID: 29786741.

4. In the section 8, supply the works of 2,4,6-Trihydroxy-3-geranyl acetophenone and chalcones. My advice is based on the following publications: 

(1) Chan YH, Liew KY, Tan JW, Shaari K, Israf DA, Tham CL. Pharmacological Properties of 2,4,6-Trihydroxy-3-Geranyl Acetophenone and the Underlying Signaling Pathways: Progress and Prospects. Front Pharmacol. 2021; 12:736339. doi: 10.3389/fphar.2021.736339. PMID: 34531753; PMCID: PMC8438195.

(2) Sousa A, Ribeiro D, Fernandes E, Freitas M. The Effect of Chalcones on the Main Sources of Reactive Species Production: Possible Therapeutic Implications in Diabetes Mellitus. Curr Med Chem. 2021; 28(8):1625-1669. doi: 10.2174/0929867327666200525010007. PMID: 32448100.
